# Is there a moderate range of impact of financialization on corporate R&D?

**Zhenghui Li**[1], **Yan Wang**[2]*

**1** Guangzhou Institute of International Finance, Guangzhou University, Guangzhou, PR China, **2** School of Economics and Statistics, Guangzhou University, Guangzhou, PR China

* 2111964052@e.gzhu.edu.cn

**Data Availability Statement:** All relevant data are within the manuscript and its Supporting information files.

**Funding:** This work was supported by the National Social Science Fund of China [grant number 19BGL050](grant held by author Zhenghui Li). The

## Abstract

How to promote corporate research and development is a particularly important issue under the background of the economy being diverted out of the real economy. By selecting samples of 1221 Chinese A-share non-financial listed companies from 2010 to 2019, this paper examines the impact of financialization on research and development through the panel threshold regression model. Then, the moderate range of the impact of financialization on corporate research and development is measured, as well as their heterogeneity is also analyzed. The research shows the following results: first, there is a dynamic relationship and moderate range between financialization and corporate research and development. Financialization has a positive effect on corporate research and development when the level of financialization exceeds 0.4748. Secondly, from further heterogeneous research, financialization has a threshold effect on research and development among enterprises with a high level of research and development. In addition, there is a promoting effect on corporate research and development only when their financialization level exceeds 0.0097 in enterprises with a high level of research and development. Therefore, in order to promote corporate research and development, financialization of non-financial enterprises should make adjustment and regulation according to the action and direction of moderate range.

## Introduction

Corporate financialization has made the economy diverted out of the real economy; at the same time, it weakened the corporate R&D. Under the impact of the economic crisis, the real economy becomes sluggish. However, as the financial industry and the real estate industry are expanding rapidly with a high rate of return, a large number of industrial funds have flowed into the financial field, which made the proportion of financial assets continue to rise driven by profit pursuit. Some scholars agreed that financial development can promote economic development. Proper financialization improve their economic value added [1]. Financial support can also improve energy efficiency in general [2], green total factor productivity [3]. However, the role of financial development in promoting economic development does not always exist [4–7]. The financialization makes capital accumulation diverted out of the real economy gradually, and their development deviated from the origin of serving the real economy, as well

funders had no role in study design, data collection and analysis, decision to publish, or preparation of the manuscript.

**Competing interests:** The authors have declared that no competing interests exist.

as being occupied by the main business [8], which leads economy to the trend of "being diverted out of the real economy". Meanwhile, unreasonable capital structure will bring bankruptcy risk to enterprises, which is not conducive to the sustainable development of enterprises [9, 10]. The real economy is the basis of steady economic development. The development of enterprise can't live without innovation, which can help to improve corporate value [11] and develop the ability of Outward Foreign Direct Investment [12]. The overall industrial development benefits from the market mechanism of resource allocation and technology diffusion, but there is still the possibility of further improvement [13]. As the micro entities, enterprises should boost spending on R&D and enhance innovation, which fundamentally promote development of the real economy. However, due to the need of investing a large amount of capital to R&D, as well as the high risk characteristics such as long R&D cycle and uncertain results [14], the short-term economic effects of enterprise innovation are not obvious. With the increasing tendency towards financial investment, more industrial capital is allocated to the financial sector, thus occupied investment on R&D, and not conducive to the development of R&D. At the same time, the profit of enterprises is more reliant on finance, which makes enterprises have more interest in finance investment, thus decreases the emphasis on R&D, and even weakens the capacities. Therefore, it is particularly important to solve the problem of how to strengthen R&D, and find the appropriate range of the impact of financialization on R&D under the background of the economy being diverted out of the real economy.

Corporate R&D is affected by financing constraints, but it is more to do with the motivation during the process of investment decision making. R&D needs long-term and stable financial support. The adequacy of funds depends on internal and external financing. Internal financing mainly relies on the capital accumulation of enterprises, while external financing depend on bank loans, bond and stock issuance, etc. However, due to the instability and limited amount of internal funds, internal financing of enterprises is difficult to meet the funds needed for R&D [15]. At the same time, as the information asymmetry, enterprises are often difficult to obtain external financing. Therefore corporate R&D will be largely affected by the degree of financing constraints [16, 17]. Except for financing constraints, corporate R&D is also affected by the motivation of enterprise investment in decision-making. In order to maintain the competitiveness and achieve sustainable development, enterprises tend to increase the capital investment on R&D and enhance innovation from the long-term development perspective in order to improve productivity, reduce production costs and improve enterprise profits. However, in the face of the decline of return rate in the real economy, enterprises allocate more funds to the financial sector in order to obtain high profits in the short term based on their profit motivation. The allocation of financial assets accelerates capital accumulation, but excessive allocation makes enterprises shift their focus to virtual economy and weakens the foundation of enterprise R&D. Therefore, profit-seeking motivation makes enterprise governance more short-sighted and weakens their R&D. Therefore, the investment of enterprises in R&D projects is different when motives of investment in decision-making are various.

Among the existing research, there is no consistent view on the impact of financialization on corporate R&D. On the one hand, financialization gives a boost on corporate R&D. Some scholars declared that asset reserve is the main motivation for enterprises to allocate financial assets. In order to deal with the uncertainty in the future, enterprises may choose to invest in financial assets which is more liquid rather than fixed assets [18]. The purpose of allocating financial assets is mainly smoothing profits, increasing the liquidity of assets and reserving assets for production and operation for enterprises. In addition, other scholars denoted that financialization can broaden the financing channels, ease the financing constraints timely, as well as offer financial support to R&D. Theurillat et al. [19] also pointed out that financialization is conducive to the spatial allocation of resources, and increases the resources used for

main business investment to a certain extent. In addition, financialization also improves the balance sheet and enhances the financing ability. Brown and Petersen [16] pointed out that enterprises facing financing difficulties generally rely on cash reserves to smooth R&D expenditure because of the high cost of adjusting R&D flow in response to short-term financing shocks.

On the other hand, financialization also has an inhibited effect on corporate R&D. With the deepening of corporate financialization, more and more researchers focus on the "crowding out" effect on R&D. Most scholars believed that financialization would crowd out the investment of real economy, which was not conducive to the development of R&D projects. Allen et al. [20] claimed that financial capital would gradually separate from industrial capital as the degree of financialization increases, which lead to decline of capital accumulation by squeezing industrial capital and profits. However, this circumstance would further reduce the investment enthusiasm of the real economy. Enterprises will allocate more financial assets, which will further increase the degree of financialization of enterprises, and the enthusiasm for investing in the real economy would be reduced ultimately. Base on the samples of non-financial companies in the United States and United Kingdom, Lazonick and Teece [21], Tori and Onaran [22] provided strong evidence that financialization would crowd out the investment of real economic. They also declared that financial activities would eventually lead to the reduction of the investment in main business, as well as squeeze the expenditure on R&D, reduce the ability of independent innovation, and thus result in the stagnation of economic development. From the motive of arbitrage investment, enterprises would over-allocate financial assets in order to pursue pure capital appreciation and realize profit maximization for shareholders. When their profit model relies more on financial assets investment, the R&D activities will be further squeezed out by financial assets. Based on research target of non-financial companies from 1994 to 2009 in South Korea, Seo et al. [23] denoted that financial assets investment is a kind of market arbitrage behavior. In order to pursue high return, enterprises would increase their investment in financial assets and purchase stocks with high return, which squeeze out the investment in R&D activities.

The impact of financialization on corporate R&D is not a linear relationship. Exploring how financialization affects corporate R&D, as well as measuring the moderate range of financialization that positively affects corporate R&D are the key questions in this paper. We chose Chinese enterprises as the study samples for the reason that Chinese real economy becomes sluggish under the impact of the economic crisis and new economic normal and the R&D capabilities of enterprises need to be strengthened. According to data, there are 1115 non-financial listed companies in China that have purchased finance products of banks and securities companies, trust loans, private equity and other financial products with 1.2 trillion RMB in 2019. Meanwhile, in the third session of the 13th National People's Congress in 2020, Premier Li Keqiang clearly pointed out in report on the work of the government that we should stimulate the vitality of market entities through reform, strengthen new drivers of development and encourage mass entrepreneurship and innovation as well. However, according to expenditure of R&D for Chinese enterprises in 2019, enterprise capital is the major source that accounting for 90.2%, which means the R&D of Chinese enterprises mainly relies on internal financing. The R&D Chinese enterprises faced with serious external financing constraints, but the high return of financial assets can play a mitigating role. All these characteristics provide a good condition for us to study the relationship between financialization and corporate R&D.

Compared with the previous studies, the main contributions of this article are as follows: First, there is a dynamic relationship between financialization and corporate R&D, and the moderate range of the level of financialization is measured. Although the current research mainly focuses on the promotion or inhibition effect of financialization on corporate R&D,

this paper tested that there is a dynamic relationship by using threshold effect model, that is, the degree and direction of the impact of financialization on R&D are different as level of financialization changes. Therefore, the impact of financialization on corporate R&D also has a moderate range. Secondly, there is a heterogeneous impact on moderate range of financialization on corporate R&D among companies with different R&D intensity. According to the different level of R&D, we divide the total sample into a high and low level. In specific, only high level enterprises have a moderate range on impact of financialization on corporate R&D. However, there is no threshold effect between financialization and R&D for enterprises with a low level of R&D.

The remainder of this paper is arranged as follows: Section 2 introduces theoretical analysis and research hypothesis, which includes the analysis of non-linear relationship and the heterogeneity of financialization and corporate R&D. Section 3 is research design, which refers to the econometric model, the measurement of each variable and its data sources. Section 4 is empirical results and analysis, including measurement results and heterogeneity test results of the influence on a moderate range between financialization and corporate R&D. Lastly, section 5 concludes our paper.

## Theoretical analysis and research hypothesis

### The dynamic relationship between financialization and corporate R&D

The motives of financialization mainly include preventive reserve and market arbitrage. The different motives of financial assets allocation have different effects on corporate R&D. The innovation activities should be supported by stable funding, but some big uncertainty, like long R&D activity cycle, low success rate and unknown commercial value, as well as high-secret measures adopted for R&D activities, making it difficult to obtain relevant information for investors, so innovation activities are subject to severe financing constraints [24]. According to the theory of financing priority, enterprises would give priority to internal financing, followed by debt financing, and finally equity financing when they are in financing needs [25]. Therefore, having sufficient internal funds is an important means to alleviate corporate financing constraints. Besides, enterprises which are subject to financial constraints would increase their current assets when their cash flow is high [26]. Compared with long-term assets, fixed assets and intangible assets, financial assets have the characteristics of stronger liquidity, lower adjustment costs, and faster realization speed. Companies can alleviate financing constraints by realizing or selling financial assets timely to when they are faced with insufficient funds [27, 28]. Therefore, in view of motivation of preventive reserves, enterprises would invest their idle assets in the financial market in order to make an asset reserve and respond to financing difficulties timely, which can ensure the continuous and stable development of R&D activities. Meanwhile the benefits from financial assets can improve business performance, as well as promote R&D. However, based on the theory of investment substitution, the financial asset investment of enterprises is a speculative arbitrage for the purpose of obtaining excess profits [29], and actually has substitution relation with entity investment [30]. By considering an investment portfolio between investments of financial asset and business operation, enterprises can implement discretionary decisions in both real economy and financial fields. When the profit rate of investment in real economy is higher than that of financial asset investment, enterprises would increase the investment in real economy and reduce financial investment. When the profit rate of investment in real economy declines, companies would reduce the scale of operating investment and increase investment in financial assets [31]. In the case of limited corporate resources, the relationship between financialization and R&D investment is a trade-off. When non-financial companies invest too much fund in finance and real estate

investment, which should have been invested in their main business, the scale of their investment in R&D would be reduced correspondingly [32]. Based on the theory of agency, both the first and the second type of agency problem may lead companies to increase their financial asset investment. Shareholders and managers are more inclined to make financial investments with a return rate far greater than that of investment in real economy, and conduct speculative arbitrage through the allocation of financial assets with the motivation of maximizing their own interests. Therefore, due to the motivation of market arbitrage, companies are encouraged to invest more assets in the financial market by great return on financial assets, which make them suppress enthusiasm to invest in fixed assets, and then their R&D capabilities would be weakened as the R&D is squeezed out by funds originally invested in R&D. When enterprises allocate financial assets, preventive reserve motives and market arbitrage motives coexist with each other, and how financialization affects enterprises R&D mainly depends on the effect of these two motives. Based on the above, we propose the following hypotheses:

Hypothesis 1: There is a dynamic relationship between financialization and enterprise R&D, namely, as the level of financialization changes, the impact degree and direction of financialization on corporate R&D are different, the level of financialization has a moderate range.

## The heterogeneous effects of different R&D intensities

With different intensity of R&D, the impact of financialization on corporate R&D activities, as well as their moderate range, are also various. First of all, the dependence on R&D activities with different R&D intensity varies greatly. It is generally believed that enterprises with high R&D intensity pay more attention to R&D activities. These enterprises often belong to the high-tech industry; their R&D is the main driving force for development. Maintaining strong technological innovation and shaping unique core competitiveness are the source of strengthening sustainable development of enterprises. At the same time, high-intensity R&D companies have higher requirements for their core technologies, operating models, and knowledge structure of employees, as well as quick-frequency updating products and technologies. They are also facing fiercer competition and need to step up the pace of R&D. However, companies with low R&D intensity lack attention to R&D, and they would allocate more financial assets under the high rate of return in the financial market, thereby further squeezing out R&D investment. Therefore, based on greater demand for R&D, companies with high R&D intensity have greater funding requirements for R&D, and they are more inclined to invest the high profits obtained from allocating financial assets into projects in order to promote the R&D. Secondly, the risks of R&D activities with different innovation strengths are diverse. Compared with low R&D intensity enterprises, high-tech R&D activities undertaken by high R&D-intensity enterprises have a longer R&D cycle and higher uncertainty, so their high-tech R&D activities often have a lower success rate. Thus, they will lose all costs invested in the early stage once the R&D activities fail. In addition, R&D is enterprises' highly confidential information, and investors always have difficulties predicting the market benefits from that. As a result, companies with high R&D intensity face more serious information asymmetry [33] and agency conflict dilemmas. It is easier to induce adverse selection behavior [34], and enterprises with high R&D intensity would face severe external financing constraints. Financialization based on the motive of preventive reserve would bring more funds to enterprises, and can also bring benefits to enterprises with high R&D intensity and relief R&D expenditures. In addition, in order to stimulate the vitality of R&D activities and promote long-term economic development, the government has implemented subsidy policies to enterprises with high R&D intensity whose R&D activities have high cost burdens. The government offers financial

support for enterprises to carry out R&D activities, share high R&D costs, and ease the problem of tight capital flows caused by large R&D expenditures [35]. However, government assistance to enterprises with low R&D intensity is relatively small, and the incentives effect for R&D activities is also low. Therefore, the negative impact of financialization on high R&D intensity enterprise is relatively small, and the phenomenon of speculative arbitrage motive for financialization squeezing R&D may be more significant in low R&D intensity companies. From the above, if the intensity of R&D is different, the inhibitory and promotion effects of financialization on their R&D activities would also be different, as well as the moderate range of financialization on corporate R&D would change accordingly. Based on this, we propose the following hypotheses:

Hypothesis 2: The moderate range of the impact of financialization on the corporate R&D varies with different R&D intensity.

## Research design

### Threshold model

Based on the dual goals of financial asset allocation, financialization may have positive or negative effects on corporate R&D. The financialization based on the motive of precautionary reserve can improve the liquidity of assets and achieve the purpose of maintaining and increasing capital value. Besides, it is also beneficial to prevent the problem of capital shortage of future investment in the main business, as well as relieving external financing constraints to a certain extent in order to promote enterprise R&D. Based on the financialization of market arbitrage motivation, non-financial enterprises without enough financial assets may lack sufficient funds for equipment upgrading and product R&D, thus the R&D may be inhibited. Therefore, there is a moderate range in the impact of financialization on corporate R&D. From the above, when studying the impact of financialization on corporate R&D, we build a panel threshold regression model from Hansen [36] as follows:

$$Inno_{it} = \beta_0 + \beta_1 I(Fin_{it} \leq \eta) + \beta_2 I(Fin_{it} > \eta) + \sum \alpha_i X_{it} + \varepsilon_{it}, \tag{1}$$

where the subscript $i$ represents an indicator for company $i$ while $t$ represents the year. $Inno$ is an independent variable that stands for the level of corporate R&D. $Fin$ is a threshold variable that represents financialization; $\eta$ is a threshold value; $I(\bullet)$ denotes index function; $X$ are the control variables, which represent the influence of other factors on corporate R&D.

This model is equivalent to a model of piecewise function as follows,

$$Inno_{it} = \begin{cases} \beta_0 + \beta_1 Fin_{it} + \sum \alpha_i X_{it} + \varepsilon_{it}, Fin_{it} \leq \eta \\ \beta_0 + \beta_2 Fin_{it} + \sum \alpha_i X_{it} + \varepsilon_{it}, Fin_{it} > \eta \end{cases}, \tag{2}$$

when $Fin_{it} \leq \eta$, where the coefficient of $Fin_{it}$ is $\beta_1$, while the coefficient of $Fin_{it}$ is $\beta_2$ when $Fin_{it} > \eta$. In addition, Eq (1) is applicable to the case of single threshold. If there is a double threshold, the model is constructed as follows:

$$Inno_{it} = \phi_0 + \phi_1 I(Fin_{it} \leq \eta_1) + \phi_2 I(\eta_1 < Fin_{it} \leq \eta_2) + \phi_3 I(Fin_{it} > \eta_2) \\ + \sum \alpha_i X_{it} + \varepsilon_{it} \tag{3}$$

where the subscript i represents an indicator for company i while t represents the year, the other remaining variables are consistent with Eq (1).

## Variables selection

The independent variable is corporate R&D, which is measured by the R&D expenditure, R&D efficiency, and R&D achievements. In view of focus on the discretion of enterprise investment behavior under the impact of the financialization, we take enterprise R&D as its investment by considering that investment may not translate into intangible assets. By comparing the ratio of net intangible assets to total assets, spending on R&D can reflect the intensity of enterprise investment much easier. Therefore, enterprise R&D in this paper is measured by the percentage of R&D expenditure in revenue.

The dependent variable is financialization, which is mainly reflected in the behavioral decision of financial assets allocation for enterprises. In view of measurement method from Demir [37], we use the share of financial assets in the total assets at the end of the period to measure the level of financialization. In this paper, the financial assets include trading financial assets, derivative financial assets, net short-term investment, net financial assets available for sale, net hold-to-maturity investments and long-term debt investments, net investment in real estate, long-term equity investments, entrusted financial management and trust products. Among them, commission financing and trust products mainly include entrusted loans, financial products and balance of investment in trust products.

Based on various influencing factors, other influencing factors of corporate R&D need to be assumed unchanged when we are studying the influence of financialization on corporate R&D, that is, other major influencing factors need to be controlled in the econometric test and set as the control variable. In the process of modeling, this paper considers adding other variables that influence technological innovation. Based on the existing literature [38–40], we take company size, company age, net cash flow of operation, net profit margin, enterprise capital intensity, capital structure and ownership concentration as control variables respectively in this paper. The specific measurement methods of variables are shown in Table 1.

## Data source and descriptive statistics

This paper selects Chinese A-share non-financial listed companies from 2010 to 2019 as the research object. In order to ensure the continuity of sample data, listed companies in the financial industry, ST (Special Treatment) and PT (Particular Transfer) listed companies, as well as companies with many data missing are removed from this sample. The reason for excluding

**Table 1. Variables description.**

| Variable type | Variable | Abbreviation | Measurement |
|---|---|---|---|
| Dependent variable | R&D | Inno | Proportion of net intangible assets in total assets |
| independent variable | Financialization | Fin | Ratio of financial assets to total assets at the end of the period |
| Control variable | Net cash flow of operation | CFO | Ratio of net cash flow of operations to total assets at the end of the period |
| | Company size | Lnsize | natural log from total assets at the end of the period |
| | Company capital intensity | Fixed | Ratio of fixed assets at the end of the period to total assets |
| | Company age | Lnage | Taking natural log that first minus the year of incorporation establishment from the current year then plus 1 |
| | Net profit margin | ROA | Ratio of net profit to total assets at the end of the period |
| | capital structure | Lev | Ratio of total liabilities at the end of the period to total assets |
| | Ownership concentration | Shrcr | Total shares proportion of the top ten shareholders |

financial listed companies is that their main business is in the financial industry, and their investment in corporate R&D is significantly different from that of non-financial listed companies, which is also inconsistent with the research objectives of this paper. As for ST and PT companies, they belong to continuous loss enterprise with poor continuing operation ability and not have the general characteristics of financial asset allocation. Based on these treatments, with the constraints of time and type, we finally choose 1221 A-share listed companies in China with a total of 12210 annual observation data as our research sample. These samples are all from China Stock Market & Accounting Research (CSMAR) database. After all the sample data are obtained, we winsorise all the continuous variables to eliminate the effect of any extreme values at 1% level.

The descriptive statistics for all variables are in Table 2. On the whole, the minimum value of Inno is 0.0000, the maximum value is 20.1500, and the average value is 2.5298, which indicates that the overall level of R&D of Chinese listed companies is not high. The minimum value of Fin is 0.0004, the maximum value is 0.6380, and the average value is 0.1026, which indicates that there is basically financialization in China's listed companies as well as Fin fluctuates greatly within the annual interval. In addition, we also conduct the sample description

**Table 2. Descriptive statistics.**

| Sample | Variable | Obs. | Mean | Std. Dev. | Min | Max |
|---|---|---|---|---|---|---|
| Full-sample | Inno | 12,210 | 2.5298 | 3.6154 | 0.0000 | 20.1500 |
| | Fin | 12,210 | 0.1026 | 0.1251 | 0.0004 | 0.6380 |
| | CFO | 12,210 | 0.0456 | 0.0676 | -0.1528 | 0.2353 |
| | Lnsize | 12,210 | 22.5460 | 1.3255 | 20.1727 | 26.4153 |
| | Lev | 12,210 | 0.4622 | 0.2037 | 0.0557 | 0.8751 |
| | Roa | 12,210 | 0.0389 | 0.0485 | -0.1645 | 0.1864 |
| | Growth | 12,210 | 0.1547 | 0.3889 | -0.7541 | 2.3429 |
| | Fixed | 12,210 | 0.2193 | 0.1712 | 0.0019 | 0.7251 |
| | Lnage | 12,210 | 2.9175 | 0.2991 | 1.9459 | 3.5264 |
| | Shrcr | 12,210 | 56.1159 | 15.5468 | 22.5200 | 90.3800 |
| Enterprise with high level of R&D | Inno | 6,100 | 2.8325 | 3.7938 | 0.0000 | 20.1500 |
| | Fin | 6,100 | 0.0888 | 0.1088 | 0.0004 | 0.6380 |
| | CFO | 6,100 | 0.0519 | 0.0627 | -0.1528 | 0.2353 |
| | Lnsize | 6,100 | 22.4670 | 1.2863 | 20.1727 | 26.4153 |
| | Lev | 6,100 | 0.4387 | 0.1908 | 0.0557 | 0.8751 |
| | Roa | 6,100 | 0.0397 | 0.0493 | -0.1645 | 0.1864 |
| | Growth | 6,100 | 0.1543 | 0.3823 | -0.7541 | 2.3429 |
| | Fixed | 6,100 | 0.2397 | 0.1504 | 0.0019 | 0.7251 |
| | Lnage | 6,100 | 2.8836 | 0.3001 | 1.9459 | 3.5264 |
| | Shrcr | 6,100 | 56.1315 | 15.5819 | 22.5200 | 90.3800 |
| Enterprise with low level of R&D | Inno | 6,110 | 2.2276 | 3.4016 | 0.0000 | 20.1500 |
| | Fin | 6,110 | 0.1165 | 0.1381 | 0.0004 | 0.6380 |
| | CFO | 6,110 | 0.0393 | 0.0717 | -0.1528 | 0.2353 |
| | Lnsize | 6,110 | 22.6248 | 1.3591 | 20.1727 | 26.4153 |
| | Lev | 6,110 | 0.4856 | 0.2133 | 0.0557 | 0.8751 |
| | Roa | 6,110 | 0.0382 | 0.0475 | -0.1645 | 0.1864 |
| | Growth | 6,110 | 0.1552 | 0.3955 | -0.7541 | 2.3429 |
| | Fixed | 6,110 | 0.1990 | 0.1875 | 0.0019 | 0.7251 |
| | Lnage | 6,110 | 2.9514 | 0.2941 | 1.9459 | 3.5264 |
| | Shrcr | 6,110 | 56.1004 | 15.5128 | 22.5200 | 90.3800 |

statistics from the intensity of R&D. The average financialization of enterprises with a high level of R&D is 0.0888, while the lower one is 0.1165. Compared with lower level ones, the average financialization of firms with high level of R&D is lower. It can be preliminarily concluded that both financialization and R&D are different among enterprises with different R&D intensity. Therefore, we ascertain that the relationship between financialization and R&D may also be heterogeneous among enterprises with different R&D intensity.

## Empirical results

### The dynamic relationship between financialization and corporate R&D

In order to test the dynamic relationship between financialization and corporate R&D, and to measure the moderate range of the impact of financialization on R&D, this paper uses the panel threshold regression model to conduct econometric tests. There are three steps: the first one is the significance test of threshold effect. We are going to check whether there is a threshold effect and what is the type of threshold model, single threshold or multiple thresholds. The second is the determination of threshold value. The third one is the parameter estimation of threshold regression. Before these, we use Bootstrap repeated sampling method to test the threshold effect, which is shown in Table 3.

From Table 3, we can see that a single threshold model passes the significance level test of 0.05, while the double threshold model does not. Therefore, there is a single threshold for the impact of financialization on corporate R&D. Then, the threshold value is determined based on the test of threshold effect. It can be seen from Table 3 that the estimated value of the threshold point is 0.4748, and the estimated value of the threshold value is within the interval from 0.4625 to 0.4882 under the confidence level of 95%.

In order to further investigate the impact of financialization on corporate R&D in the interval formed by different thresholds, the panel double threshold model is adopted to conduct parameter estimation based on the threshold value. Here are the parameter results in Table 4.

The influence of financialization on corporate R&D is dynamic, and the level of financialization has a moderate range. From Table 4, financialization has a significant inhibiting effect on R&D when Fin≤0.4748 with an estimated coefficient of -0.7661. Financialization has a significant promoting effect on R&D when Fin> 0.4748 with a coefficient of 0.9250. The relationship between financialization and corporate R&D is dynamic. To be specific, the financialization first negatively affects the corporate R&D, and then has a positive impact when financialization reaches a certain turning point. There are two reasons for this. On the one hand, when financialization is at a low level, enterprises tend to be more inclined to invest in financial markets, suspend R&D activities, or allocate financial assets depending on market arbitrage motives because benefits from the financial assets allocation cannot meet the capital requirements of innovation activities. On the other hand, as the high rate of return in the financial sector makes corporate investment myopic, enterprises allocate more on financial assets while ignoring the investment for enterprises' entities in pursuit of capital appreciation, thus squeezing out the investment in corporate R&D. Therefore, when the financialization is at a low level, the increase of financial assets has a "crowding out" effect on corporate R&D. However, when the financialization reaches a certain level, these measures, such as: enhancing

**Table 3. Test result of threshold effect.**

| Threshold model | F-value | P-value | Threshold type | Threshold value | Confidence interval |
|---|---|---|---|---|---|
| Single threshold | 35.1 | 0.01 | Single threshold | 0.4748 | (0.4625, 0.4882) |
| Double threshold | 14.02 | 0.1867 | | | |

**Table 4. Regression results of threshold effect.**

| Variable | Inno |
|---|---|
| Fin≤0.4748 | -0.7661*** |
| | (0.2852) |
| Fin>0.4748 | 0.9250*** |
| | (0.3081) |
| CFO | 0.1477 |
| | (0.3049) |
| Lnsize | 0.1622*** |
| | (0.0480) |
| Lev | -2.1289*** |
| | (0.1972) |
| Lnage | 3.2210*** |
| | (0.1514) |
| Fixed | 0.5704** |
| | (0.2458) |
| Roa | -7.6094*** |
| | (0.4924) |
| Shrcr | -0.0033 |
| | (0.0023) |
| Constant | -9.1370*** |
| | (0.7931) |
| Observations | 12,210 |
| Number of firmcode | 1,221 |
| R-squared | 0.1491 |

**Note**: Robust standard errors in parentheses.

*** p<0.01,

** p<0.05,

* p<0.1

of funds liquidity from efficient returns of financial assets, easing financing constraints from R&D activities, supporting funds for R&D, prompting investment in R&D, can make a promotion to corporate R&D. Therefore, the increase of financial assets can be acted as a kind of promoting effect on enterprise R&D when financialization reaches a certain level. Based on this, there is a moderate range on the impact of financialization on corporate R&D; that is, financialization has a promoting effect on corporate R&D when the level of financialization exceeds 0.4748.

## The heterogeneous effects of different R&D intensities

From the above empirical results, we can understand the impact of financialization on corporate R&D on the whole, and obtain the moderate range of financialization as well. However, heterogeneous effects are widely investigated in the study of financial issues [41–43], the effects of financialization are different among enterprises with different R&D intensity. Therefore, this paper further studies the heterogeneity of relationship between financialization and corporate R&D among enterprises with different R&D intensity. We first rank the R&D level of each enterprise from small to large according to the annual mean value. The enterprises in the top 50% are enterprises with high R&D level, and the rest 50% are enterprises with low level of R&D by the median dividing.

**Table 5. Test result of threshold effect.**

| Enterprise types | Threshold model | F-value | P-value | Threshold type | Threshold value | Confidence interval |
|---|---|---|---|---|---|---|
| Enterprise with high level of R&D | single threshold | 34.72 | 0.0000 | single threshold | 0.0097 | (0.0094, 0.0099) |
| | Double threshold | 7.07 | 0.5600 | | | |
| Enterprise with low level of R&D | single threshold | 14.41 | 0.2267 | | | |

From Table 5, we can see that a single threshold model passes the significance level test of 0.05, while the double threshold model does not. Therefore, for enterprises with high R&D level, there is a single threshold for financialization. While, there is no threshold effect on financialization for firms with low R&D level because the single threshold model does not pass the significance test of the model. On the basis of the threshold effect test, the threshold value of financialization with high R&D level is determined. It can be seen from Table 5 that the estimated value of the threshold point is 0.0097. Meanwhile the estimated value of the interval of the threshold value is from 0.0094 to 0.0099 under the confidence level of 95%.

In order to further investigate the impact of financialization with high R&D level on corporate R&D in the interval formed by different thresholds, the panel single threshold model is adopted to conduct parameter estimation in Table 6.

**Table 6. Regression result of threshold effect of enterprises with high R&D level.**

| Variable | High level of R&D |
|---|---|
| | Inno |
| Fin≤0.0097 | -1.7730*** |
| | (0.5179) |
| Fin>0.0097 | 1.6571*** |
| | (0.5950) |
| CFO | 0.1190 |
| | (0.6442) |
| Lnsize | 0.1181 |
| | (0.0912) |
| Lev | -4.0015*** |
| | (0.3570) |
| Lnage | 5.8457*** |
| | (0.2967) |
| Fixed | 0.5931 |
| | (0.5045) |
| Roa | -12.0146*** |
| | (0.8672) |
| Shrcr | 0.0201*** |
| | (0.0045) |
| Constant | -13.7356*** |
| | (1.5064) |
| Observations | 6,100 |
| R-squared | 0.2170 |
| Number of firmcode | 610 |

**Note**: Robust standard errors in parentheses.

*** $p < 0.01$,

** $p < 0.05$,

* $p < 0.1$

Among firms with different R&D intensity, the moderate range of the impact of financialization on R&D is different. From Table 6, for enterprises with high R&D level, financialization has a significant inhibiting effect on R&D when Fin≤0.0097 with an estimated coefficient of -1.7730. When Fin> was 0.0097, financialization has a promoting effect on R&D with the regression coefficient of 1.6571. The dynamic characteristic of the relationship between financialization and corporate R&D is more significant. However, for enterprises with low R&D level, there is no threshold effect on the impact of financialization on R&D, and there is no moderate range for financialization. Referring to the reasons, enterprises with different strengths of innovation may have different dependence and initiative of R&D. Enterprises with high levels of innovation pay more attention to R&D, and they also conduct innovation activities with higher technologies. Thus, they would face more serious financing constraints with a greater risk of failure on research and development activities. The development of financial markets offers great help to the enterprise with high innovation levels, just as [44] study, they found countries with relatively developed stock markets, as well as some industries that are highly dependent on external financing and are high-tech intensity, showed higher levels of innovation. Enterprises with a high level of R&D are more inclined to invest the income from financial assets in R&D activities. At the same time, the government also gives policy support to them, which provides a good foundation for their R&D. Therefore, the effects of financialization on corporate R&D will be different among enterprises with different R&D intensity. Besides, the empirical results also show that there is a moderate range on the impact of financialization on R&D for enterprises with a high level of R&D. That is, financialization has a promoting effect on corporate R&D only when the level of financialization exceeds 0.0097. Compared with empirical results of the full sample in Table 4, the threshold of financialization of enterprises with high R&D level is smaller. That is, a lower level of financialization is required to have a positive impact on R&D among enterprises with a high level of R&D. It further indicates that high R&D level has greater demand for R&D investment, and their financial assets allocation is more motivated by asset reserve, as well as financialization plays more significant role in promoting R&D.

## Conclusion and suggestion

By selecting 1221 Chinese A-share non-financial listed companies from 2010 to 2019 as sample, this paper analyzes the impact of financialization on corporate R&D through the panel threshold regression model, and draws the following conclusions:

First, the influence of financialization on corporate R&D is dynamic, and the level of financialization has a moderate range. Financialization has a significant inhibiting effect on R&D when Fin ≤0.4748. Financialization promotes the R&D when Fin > 0.4748. In other words, financialization has a promoting effect on R&D only when its level exceeds 0.4748. When financialization is at a low level, if the profit from financial asset allocation cannot meet the requirements of capital needs of R&D activities, enterprises are more inclined to invest in the financial market, postpone R&D activities, and allocate financial assets out of market arbitrage motive. When financialization reaches a certain level, the efficient returns brought by financial assets would enhance funds liquidity, guarantee funds supply for R&D, and encourage enterprises to invest in R&D. Thus, the increase of financial assets can promote R&D for enterprises.

Second, further research shows that moderate range of the impact of financialization on corporate R&D is heterogeneous among firms with different R&D intensity. Among firms with higher levels, financialization has a threshold effect on R&D. Among firms with lower levels, financialization has no threshold effect. In enterprises with higher levels of R&D intensity,

the impact of financialization on R&D has a moderate range, and their threshold value is smaller than that in the full sample. Financialization has a promoting effect on enterprise R&D when financialization level exceeds 0.0097.

From the above conclusions, some relevant suggestions from non-financial enterprises, financial institutions and government are provided as follows: to begin with, there is a dynamic relationship between financialization and corporate R&D, and the increase of financial assets has a promoting effect on firm's R&D only when financialization reaches a certain level. However, with the increase of financial assets, their negative effects would also increase, such as "crowding out" the real economy [22] and exacerbating the build-up of asset price bubbles [45]. Based on this, non-financial enterprises should put their R&D projects at an important strategic position, as well as balancing the proportion of financial assets and investment in R&D, avoiding excessive financialization. Thus, their financial investment can not only promote enterprises development but also avoid financial risks. Then, for the financial sector, the following ways can be suggested, for example, playing a proper role in serving the real economy by providing diversified and innovative financial products to the real economy. At the same time, it should perfect the information disclosure platform of R&D-related investment projects, which can alleviate the increase in financing constraints caused by information asymmetry, and then provide convenience for corporate innovation and financing. Finally, as for the government, firstly, they should pay attention to the impact of capital flows on China's financial cycle spillovers [46], as well as the impact of economic policy uncertainty on financial market volatility [47]. The government should strengthen the market-oriented reform of the financial system and improve the development of the financial market in order to offer favorable conditions for enterprises to invest in R&D. At the same time, the financial supervision system should be improved to prevent excessive financialization. Secondly, improving the social financing system, optimizing financing channels as well as easing the financing constraints are also deserved to have a try. Thirdly, they should encourage non-financial enterprises to conduct R&D activities, provide relevant preferential policies for enterprises, and support R&D projects of real enterprises vigorously.

## Limitations and prospects

In light of the above findings, it is important to note some limitations of this study. First of all, it is insufficient in the measurement of variables. This research measures the corporate financialization only from the share of financial assets held, which may not fully measure the level of corporate financialization. Future research needs to continuously optimize the measurement of corporate financialization. For example, it can be considered to measure corporate financialization from multiple levels simultaneously. Secondly, when studying the moderate range of impact of corporate financialization on corporate R&D, this paper is limited to the perspective of micro enterprises, without considering the impact of the macroeconomic environment, however, it cannot be ignored the impact of macroeconomic fluctuations on the behavior of micro enterprises. Future research can consider how the moderate range of impact of corporate financialization on corporate R&D will change under the influence of economic policy uncertainties.

## Supporting information

**S1 Data.**
(XLSX)

## Author Contributions

**Conceptualization:** Zhenghui Li, Yan Wang.

**Data curation:** Yan Wang.

**Formal analysis:** Zhenghui Li, Yan Wang.

**Funding acquisition:** Zhenghui Li.

**Investigation:** Zhenghui Li, Yan Wang.

**Methodology:** Zhenghui Li, Yan Wang.

**Project administration:** Zhenghui Li.

**Resources:** Yan Wang.

**Software:** Yan Wang.

**Supervision:** Zhenghui Li, Yan Wang.

**Visualization:** Yan Wang.

**Writing – original draft:** Yan Wang.

**Writing – review & editing:** Zhenghui Li.

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
