## [Decision Letter · Decision Letter 0]

21 May 2021

PONE-D-21-12838

Is there a moderate range of impact of  financialization on corporate R&D?

PLOS ONE

Dear Dr. Yan Wang,

Thank you for submitting your manuscript to PLOS ONE. After careful consideration, we feel that it has merit but does not fully meet PLOS ONE’s publication criteria as it currently stands. Therefore, we invite you to submit a revised version of the manuscript that addresses the points raised during the review process.

We look forward to receiving your revised manuscript.

Kind regards,

László VASA, PhD

Academic Editor

PLOS ONE

Journal Requirements:

[The authors would like to thank Guangzhou University for sponsoring this research.]

 [This work was supported by the National Social Science Fund of China [grant number 19BGL050](grant held by author Zhenghui Li). The funders had no role in study design, data collection and analysis, decision to publish, or preparation of the manuscript.]

Reviewers' comments:

Reviewer's Responses to Questions

**Comments to the Author**

1. Is the manuscript technically sound, and do the data support the conclusions?

Reviewer #1: Yes

Reviewer #2: Yes

2. Has the statistical analysis been performed appropriately and rigorously? 

Reviewer #1: Yes

Reviewer #2: Yes

3. Have the authors made all data underlying the findings in their manuscript fully available?

Reviewer #1: Yes

Reviewer #2: Yes

4. Is the manuscript presented in an intelligible fashion and written in standard English?

Reviewer #1: Yes

Reviewer #2: Yes

5. Review Comments to the Author

Reviewer #1: Paper seems well drafted and researched.

Though authors are does not seem native English speakers, language seems good.

I have a few concerns: What was the basis of selection of specific financial variables? Was there is possibility of many other financial variables? What is rational?

What is use of this research to Industry and academia?

What are future directions of research?

Verify calculations for correctness and verify panel threshold regression model of Hansen and restated model. Do all calculations and remove errors if any.

I will advise you improve literature or background study. Refer a few articles to improve, https://doi.org/10.21314/JOP.2020.245 and http://dx.doi.org/10.1504/IJMFA.2018.095976

Revise and resubmit paper.

Reviewer #2: The paper investigates a widely researched topic from a new aspect, how the financialization impatcs on corporate R&D. The literature is well processed and analyzed. A (general) part of the literature review is included in the Introduction (without formulationg a subchapter), the other part is to find in the theoratical foundation (second chapter). The sources are analysed well, in a critical and comprehensive way.

The authors selected the appropriate statistical methodology toolset (panel threshold regression model ) and applied well. The results are logical, supported by the methodology and datasets. The conclusions are based on the results, reflecting to the question set in the paper's title.

As a summary, in my opinion it is an excellent academic paper. My only suggestion is to include the limitations of the research at the end of the paper.

6. PLOS authors have the option to publish the peer review history of their article (what does this mean?). If published, this will include your full peer review and any attached files.

Reviewer #1: No

Reviewer #2: No

---

## [Author Response · Author response to Decision Letter 0]

1 Jun 2021

Reply to the Reviewer 1：

1、What was the basis of selection of specific financial variables? Was there is possibility of many other financial variables? What is rational?

Reply: Thanks for your concerns and comments. According to Accounting Standards for Business Enterprises, enterprise assets can be divided into operational assets and financial assets. Financial assets are mainly used for investment and financing activities of enterprises. Therefore, when measuring corporate financialization, financial assets items and quasi-financial assets items in the balance sheet of enterprises should be selected. The selection of financial variables in the control variables is mainly based on the relationship between variables and R&D. Although there are many factors influencing R&D, we choose financial variables that have a significant impact on R&D according to relevant references [38-40].

References:

38. He JJ, Tian X. The dark side of analyst coverage: The case of innovation. J Financ Econ. 2013;109(3):856-78. doi: https://doi.org/10.1016/j.jfineco.2013.04.001.

39. Hong M, Drakeford B, Zhang K. The impact of mandatory CSR disclosure on green innovation: evidence from China. Green Finance. 2020;2(3):302-22. doi: https://doi.org/10.3934/GF.2020017.

40. Su K, Liu H. Financialization of manufacturing companies and corporate innovation: Lessons from an emerging economy. Manag Decis Econ. 2021. doi: https://doi.org/10.1002/mde.3278

2、What is use of this research to Industry and academia?

Reply：Thanks for your comments. To Industry, this research provides a reference for enterprises to allocate financial assets rationally, and provides an appropriate range of the proportion of corporate financial assets held by enterprises, which promotes R&D, and achieves sustainable development goals. To academia，the current research mainly focuses on the promotion or inhibition effect of financialization on corporate R&D, this research provides a theoretical basis for the dynamic relationship between financialization and R&D, and provides empirical evidence for the moderate range of the impact of financialization on R&D.

3、What are future directions of research?

Reply: Thanks for your comments. We have limitations in the measurement of financialization, and the relationship between financialization and R&D needs to be further studied. Future research can optimize the measurement of financialization and study the moderate range of the impact of corporate financialization on R&D under the influence of macroeconomic environment. For specific explanation, the research prospect is added at the end of this paper (Line 526-529, Line 533-535). The content of the modification has been marked in red in the article.

4、Verify calculations for correctness and verify panel threshold regression model of Hansen and restated model. Do all calculations and remove errors if any.

Reply: Thanks for your concerns and comments. To ensure the accuracy of the results in this article, we rechecked the original data, the calculated data, the STATA procedures we used and reworked the results. After checking, there is no calculation error in this paper, and the result of panel threshold regression model of Hansen is correct.

5、I will advise you improve literature or background study. Refer a few articles to improve, https://doi.org/10.21314/JOP.2020.245 and http://dx.doi.org/10.1504/IJMFA.2018.095976

Reply: Thanks for your advice. In order to fully explain the significance of our research, we add the negative impact of financial development, the importance of corporate R&D, and the financing constraints faced by corporate R&D (line 47-52, line 73-77). We also added references (reference 3, 9, 10, 11, 12, 15, 16, 17).The content of the modification has been marked in red in the article.

Reply to the Reviewer 2：

1、My only suggestion is to include the limitations of the research at the end of the paper.

Reply: Thanks for your comments. We have limitations in the measurement of financialization, and the relationship between financialization and R&D needs to be further studied. For specific explanation, the research limitations are added at the end of this paper (Line 523-526, Line 529-533). The content of the modification has been marked in red in the article.

---

## [Editor Report · Decision Letter 1]

4 Jun 2021

Is there a moderate range of impact of  financialization on corporate R&D?

PONE-D-21-12838R1

Dear Yan Wang,

We’re pleased to inform you that your manuscript has been judged scientifically suitable for publication and will be formally accepted for publication once it meets all outstanding technical requirements.

Kind regards,

Prof. László Vasa, PhD

Academic Editor

PLOS ONE
---

## [Editor Report · Acceptance letter]

21 Jun 2021

PONE-D-21-12838R1 

Is there a moderate range of impact of financialization on corporate R&D? 

Dear Dr. Wang:

I'm pleased to inform you that your manuscript has been deemed suitable for publication in PLOS ONE. Congratulations! Your manuscript is now with our production department. 

Kind regards, 

on behalf of

Prof. Dr. László Vasa 

Academic Editor

PLOS ONE